# Validation of Satellite and Merged Rainfall Data over Ghana, West Africa

**Winifred Ayinpogbilla Atiah *,† , Leonard Kofitse Amekudzi † , Jeffrey Nii Armah Aryee † and Kwasi Preko † and Sylvester Kojo Danuor †**

Meteorology and Climate Science Unit, Department of Physics, Kwame Nkrumah University of Science and Technology (KNUST), UPO-PMB Kumasi, Ghana; leonard.amekudzi@gmail.com (L.K.A.); jeff.jay8845@gmail.com (J.N.A.A.); kwasipreko.sci@knust.edu.gh (K.P.); skdanuor.cos@knust.edu.gh (S.K.D.)
* Correspondence: winifred.a.atiah@aims-senegal.org
† These authors contributed equally to this work.

**Abstract:** In regions of sparse gauge networks, satellite rainfall products are mostly used as surrogate measurements for various rainfall impact studies. Their potential to complement rain gauge measurements is influenced by the uncertainties associated with them. This study evaluates the performance of satellites and merged rainfall products over Ghana in order to provide information on the consistency and reliability of such products. Satellite products were validated with gridded rain gauge data from the Ghana Meteorological Agency (GMet) on various time scales. It was observed that the performance of the products in the country are mostly scale and location dependent. In addition, most of the products showed relatively good skills on the seasonal scale (r > 0.90) rather than the annual, and, after removal of seasonality from the datasets, except ARC2 that had larger biases in most cases. Again, all products captured the onsets, cessations, and spells countrywide and in the four agro-ecological zones. However, CHIRPS particularly revealed a better skill on both seasonal and annual scales countrywide. The products were not affected by the number of gauge stations within a grid cell in the Forest and Transition zones. This study, therefore, recommends all products except ARC2 for climate impact studies over the region.

**Keywords:** validation; satellite; merged products; rainfall; Ghana

## 1. Introduction

Rainfall plays an essential role in the effective management of key socio-economic activities including, rain-fed agriculture, hydro-electrical power generation, and water resources [1–3]. This applies to Ghana and the rest of the West African countries in general like few other regions in the world, where rain-fed agriculture is predominant [4,5]. The latter was strongly impacted by both the dearth and excess of rainfall during the severe Sahelian drought in the 1970s–1980s and several flood seasons in the last decade, respectively, which have raised awareness about the significant vulnerability of the West African region. Our understanding of these features has greatly improved by evaluating long-term rainfall recordings from various rain gauge stations all over West Africa [6,7] since they provide direct and therefore precise rainfall estimates. In recent decades, however, inadequate resource provision and management of meteorological agencies in the region have led to fast deterioration of rain gauge networks [8,9] as well as a more difficult accessibility [10]. The latter is further complicated by the poor coverage of rainfall data in the Global Telecommunication System (GTS) [6], which have generally caused limited confidence in recent rainfall trends for West Africa [11].

Technological progress and the desire to overcome the general data paucity in the tropics have led to a series of satellite-based rainfall products, which have been designed for different

applications on various spatio-temporal scales [12,13]. The underlying retrieval methodologies of satellite-based products can be classified into infrared (IR) and microwave-based (MW) techniques. While the IR-based approach benefits from continuous rainfall monitoring on geostationary satellites throughout the day, it falls short in accuracy due to the highly nonlinear relationship between cloud-top temperature and rainfall intensity [14]. MW-based techniques, on the other hand, relate their rainfall estimates directly to precipitation droplets within clouds, and operates currently only on low-earth orbiting (LEO) satellites, leading to a substantially lower temporal sampling over a given region [15]. Therefore, efforts have been made to merge their capabilities and to compensate for the deficiencies of IR- and MW-based rainfall estimation. Further improvements include the incorporation of rain gauge data to calibrate the satellite retrievals to specific regions, which, for instance, was operationally performed in the Tropical Rainfall Measuring Mission (TRMM) Multi-Satellite Precipitation Analysis (TMPA) in Cattani et al. [14]. The increasing usage of satellite-based rainfall data in recent times, the necessity of quality feedback for developers, and end-users through validation with "ground truth" data has risen [16]. This also includes gridded, gauge-only rainfall datasets, such as the Global Precipitation Climatology Center (GPCC) [17], which enable pixel-to-pixel validations apart from the general point-to-pixel approach, but whose quality is strongly dependent on the availability of rain gauge stations [17,18].

For West Africa, a number of studies performed an inter-comparison of the various rainfall products in order to assess the quality of the datasets. Although it is difficult to directly compare these studies because of varying methodologies, reference datasets, and region of interests, there is some general consensus on its performance. Some of these studies report on seasonal to decadal differences [16,19,20], an underestimation of heavy rainfall [19,21], and a lower bias in gauge-calibrated than in non-calibrated satellite-based datasets [16,18]. Furthermore, Dembélé and Zwart [19] provided specific application-related recommendations of the datasets based, among others on their tendency to under- and over- estimate rainfall events, which and desired for drought and flood monitoring, respectively. As such, namely PERSIANN, CHIRPS, and TRMM 3B42 [22–24] are proposed to be used for flood forecasting, whereas the other evaluated datasets ARC, RFE and TARCAT [12,13,25,26] are recommended for drought monitoring. However, validation studies over Ghana are limited. The very few studies carried out include, for example, Manzanas et al. [4] and Amekudzi et al. [27]. In recent times, Aryee et al. [2] developed a gridded rainfall dataset (GMet v1.0) based on 113 GMet stations in a 23-year period (1990–2012), which is unique within the Guinea Coast region. Thus, by using the novel GMet v1.0 dataset of Aryee et al. [2], the present study extensively validates the performance of two gauge-only and six satellite and merged rainfall products over Ghana during the TRMM era between 1998–2012, which is further regionalized in four prevailing agro-ecological zones. The remaining parts of the study is structured as follows: Section 2 presents the study area while the data used in the study is described in Section 3. The methodology employed is given in Section 4. The results are discussed in Section 5 and conclusions are provided in Section 6.

## 2. Study Area

Ghana is located in the Guinea Coast of West Africa and its climate is monsoonal. Rainfall in the Southern part of the country is bi-modal and uni-modal in the semi-arid north. On the coast, the major rainy season has its peak in May–June and is much higher than the peak of the minor rainy season, which is in October to early November [28]. In between, the little dry season culminates in August, with the major dry season occurring in boreal winter. Farther inland, the rainy season is still bi-modal, but the peak of the minor rainy season is much higher and occurs in September–October. The northern part of the country is characterized by a uni-modal rainy season similar to the Sahelian region but has its peak in August [28].

For the purpose of forecast interpretation and reporting, the country has been divided into four agro-ecological zones by GMet, which favors natural vegetation distribution, namely, the Coastal, Forest, Transition, and Savannah zones [2,3,27]. Recently, Maranan et al. [29] have shown that,

although the bulk of rainfall is produced from Mesoscale Convective Systems (MCSs), less-organized and more isolated systems become more important on the coast. In a 100 km wide strip paralleling the coast, land–sea breeze convection becomes a relevant factor for rainfall producing systems. This led us to slightly modify the zones for the present study. Hence, in this study, the Coastal zone extends to the west of the Cape Three Point where naturally dense rain Forests would be prevalent (see Figure 1b). Otherwise, the other three classical zones remain the same and were approximated using 0.5° × 0.5° squares (Figure 1b).

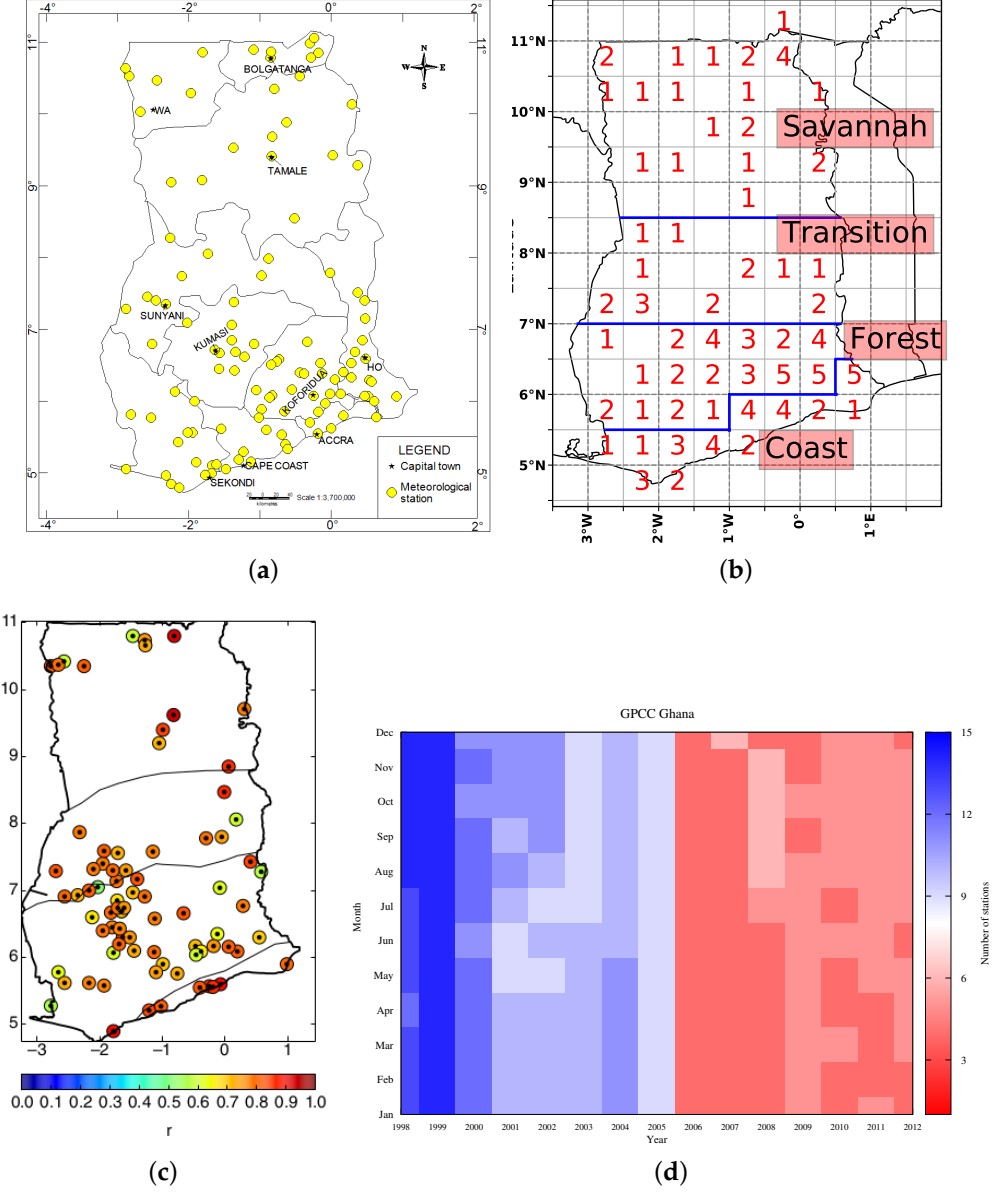

**Figure 1.** Maps of Ghana indicating the locations of all the 113 rain gauge stations over the entire country (**a**), approximate locations of the four ecological zones (blue lines showing boundaries) in Ghana with the number of gauge stations per grid-cell (**b**), point-pixel rainfall validation showing correlation coefficients with statistical significance at 99% confidence level shown as black dots over all four agro-ecological zones (**c**) (adapted from our previous paper Aryee et al. [2]), and the gauge number distribution over Ghana for the period of 1998–2012 from GPCC (**d**).

## 3. Data Source

### 3.1. Gauge Data

Monthly rainfall data were obtained from a relatively dense network of 113 GMet rain gauge stations for a period of 15 years (1998–2012). The choice of the study period was chosen to coincide with the TRMM era and, within this period, less than 5% of data gaps were present in the station data. The GMet rain gauge stations are unevenly spread over Ghana with the southern half of the country relatively dense compared to the northern half (see Figure 1a).

The minimum surface curvature with tensioning (MSC) was used to grid the 113 gauge dataset as described in Aryee et al. [2]. The gridding procedure works in similitude to a slender and linearly-elastic plate moved through data values with infinitesimal bending, employing a surface of relatively small total squared curvature, as well as continuous second derivatives. In simple terms, it is like passing a string through the gauge points and then vibrating the string so that the vibrations vary along the string points. However, to reduce spikes and "wobbles", a set tension is applied within the string to regulate the vibrations and limit inflection of data points. The MSC algorithm works with similar governing principle. The limitation, however, as described in Aryee et al. [2], is that the method struggles with extrapolations. Therefore, a common issue that may result from GMetv1.0 is extraneous values along the boundaries because outer boundary data beyond the country were unavailable for use in the data gridding process. However, arguably, the method was found to interpolate better, with details on performance provided in our previous paper, Aryee et al. [2].

To evaluate the reproducibility of GMet v1.0, relative to gauge, a point-pixel validation was carried out. Stations that were earlier discarded, due to the presence of a continuous and long gap in data, were used in validating GMet v1.0. Data used for validation were sampled from stations that, although having continuous gaps greater than 10% and as such was not included in the gridding, still had high rainfall data to gap ratio. In all, a total of 78 stations meeting these criteria were used for the point-pixel validation. For each of the 78 stations validated, we select data from the nearest grid box in GMet v1.0 that each station is bounded by and validate with gauge data. As shown in Figure 1c, point-pixel correlation coefficients were on average greater than 0.7, and statistically significant at a 99% confidence level. This indicates at least a 70% probability that rainfall over the country has been adequately reproduced by GMet v1.0. Moreover, four out of the 78 sampled stations were found to have correlations above 0.9, with two of them lying in the upper Savannah, one in the southwest Forest zone and the other in the Coast [2].

### 3.2. Validated Rainfall Products (VRPs)

The Global Precipitation Climatology Project (GPCP) is a global multi-satellite rainfall product whose algorithm combines precipitation data from rain gauge stations, satellites, and sounding observations to estimate rainfall [30]. Microwave estimates are used to adjust infra-red estimates [31]. GPCP has a spatial resolution of $2.5° \times 2.5°$. The Tropical Measuring Mission (TRMM) algorithm incorporates data from other satellite products which involves the use of passive microwave sensors (Precipitation Radar (PR) and TRMM Microwave Imager (TMI)) and Infrared measurements [18]. The TRMM precipitation data are adjusted with gauge data on monthly time scales. TRMM is a multi-satellite precipitation rainfall product which provides data on spatial resolution of $0.25° \times 0.25°$ and 3 hourly, daily, and monthly temporal resolutions over the Tropics. Global Precipitation Climatology Centre (GPCC) is a gauge-only rainfall product which provides global analysis of monthly precipitation based on in-situ rain gauge data at various spatial resolutions. GPCC incorporates quality-controlled data from nearly 67, 200 stations worldwide and has a spatial resolution of $0.5° \times 0.5°$ [17,32,33]. The Climate Research Unit (CRU) is based on the reanalysis of 4000 stations spanning a period of 1901–2012 [34]. The CRU precipitation product provides information on the number of stations that were used for the interpolation. This enables determination of the reliability

of rainfall records [35]. Global monthly precipitation fields from the CRU product are in a spatial resolution of 0.5° × 0.5°.

The Climate Prediction Center (CPC) Merged Analysis of Precipitation (CMAP) applies a merging technique to combine gauge data and satellite estimates using the inverse distance weighting directional shadowing algorithm [36,37]. CMAP is a global precipitation product with a monthly temporal resolution and a spatial resolution of 2.5° × 2.5° for a climatological period of 1979–present [37,38]. The Tropical Applications of Meteorology using Satellite data and ground-based observations (TAMSAT) is based on highly resolved thermal Infra-red observations from 1983–present [13]. The TAMSAT algorithm assumes cold cloud-top temperatures of tropical storms and identifies clouds that and rain bearing [39]. The Cold Cloud-top temperatures and obtained from the METEOSAT Infra-red imageries. The Cold Cloud Duration (CCD) imageries are produced by summing over ten days the time length at which a satellite pixel is colder than a particular threshold temperature. The TAMSAT algorithm assumes that the precipitation amount is directly proportional to the CCD and each parameter of the linear relationship is obtained using gauge data. Nonetheless, the algorithm is not impacted by changes in the availability of ground stations used for the calibration. TAMSAT rainfall estimates come in spatial resolution of 4 km × 4 km over Africa. The Climate Hazards Group Infra-red Precipitation with Station data (CHIRPS) is a combination of satellite imageries with resolution of 0.05° × 0.05° and in-situ station data [5,40]. CHIRPS algorithm incorporates various temporal resolutions of Cold Cloud Duration (CCD) based precipitation. CHIRPS is quasi-global with daily temporal resolution and spatial resolution of 0.25° × 0.25° spanning a climatological period of 1981–near present [5,26]. The precipitation measurement algorithm of African rainfall climatology (ARC2) includes estimates from IR data of METEOSAT, PM sensors, and daily rainfall data from GTS reports [25,41]. ARC2 rainfall data have spatial resolution of 0.1° × 0.1° and daily temporal resolution. Summary of the information on the validated rainfall products is provided in Table 1.

**Table 1.** Summarized information on the eight validated rainfall products

| Dataset | Spatial Resolution [°] | | Temporal Resolution | | Product Type | Coverage | References |
|---|---|---|---|---|---|---|---|
| | Original | Used | Original | Used | | | |
| CMAP | 2.50 | 2.50 | Monthly | Monthly | Satellites + gauge | Global | Khan et al. [37] |
| GPCP V2.2 | 2.50 | 2.50 | Monthly | Monthly | Satellites + gauge | Global | Adler et al. [30] |
| GPCC V6 | 0.50 | 0.50 | Monthly | Monthly | Gauge-only | Global | Schneider et al. [17] |
| ARC2 | 0.10 | 0.50 | Daily | Monthly | Satellites + gauge | Africa | Zhang et al. [41] |
| CRU TS3.21 | 0.50 | 0.50 | Monthly | Monthly | Gauge-only | Global | Harris et al. [35] |
| TAMSAT V3 | 0.0375 | 0.50 | Daily | Monthly | Satellites + gauge | Africa | Seyama et al. [39] |
| TRMM 3B43 | 0.25 | 0.50 | Monthly | Monthly | Satellites + gauge | Tropics | Schneider et al. [17] |
| CHIRPS V2 | 0.25 | 0.50 | Daily | Monthly | Satellites + gauge | Africa | Atiah et al. [3] |

## 4. Validation Methodology

This validation study used monthly gridded rainfall data of spatial resolution, 0.5° × 0.5° as described in Aryee et al. [2] to assess the performance of eight rainfall products over Ghana. Thereafter, the gridded gauge data was re-aggregated to the respective spatial resolutions of the validated rainfall products (see Table 1) for an unbiased evaluation. The validation is carried out based on two main strategies, namely, countrywide and inter-zonal assessments (see Figure 1b). Rainfall data was extracted from the gridded gauge, TRMM, GPCC, CRU, ARC2, TAMSAT and CHIRPS products for the four ecological zones. Due to the coarse spatial resolutions of CMAP and GPCP, they were excluded from the inter-zonal assessments. Furthermore, the quality of all the rainfall products were assessed with a suite of statistical techniques adapted from Amekudzi et al. [27]. These are the Pearson correlation coefficient, the Root-Mean-Square Error, bias, and efficiency. The Pearson correlation coefficient (r) measures the linear relationship between the satellite and the gauge estimates with range between −1–+1. Root-Mean-Square Error (RMSE) measures the mean deviation of the estimates from the observations. The Efficiency (Eff) measures the quality of the satellite products to estimate the observations. Bias is a measure of the extent to which satellite under/over estimates gauge

observations. Centered root mean square error (CRMSE) is a statistical metric which measures the magnitude of random error. The VRPs were further evaluated after removal of seasonality from the datasets. The seasonality in datasets were removed by the method of differencing. First, monthly means of the rainfall data for the period of study were computed. Thereafter, each month's mean value was subtracted from its respective month data throughout the period of study. For example, the mean value for January is subtracted from all January rainfall data for the entire period and so on to the last month. Lastly, Taylor and error diagrams were constructed with the aid of the Verif 1.0.0 software package (developed by Thomas Nipen, David Siuta, and Tim Chui) in order to further statistically assess the general performance of the VRPs countrywide and in the four zones. The mathematical representations of the statistical metrics used are given in Equations (1)–(4):

$$r \;=\; \frac{N \sum_{i=0}^{N} G_i S_i - \sum(G_i) \sum(S_i)}{\sqrt{(N \sum G_i^2 - \sum(G_i^2))((N \sum S_i^2 - \sum(S_i^2))}} \tag{1}$$

$$Bias \;=\; \frac{\sum_{i=0}^{n} S_i}{\sum_{i=0}^{n} G_i} \tag{2}$$

$$RMSE = \frac{\sqrt{\frac{1}{n} \sum_{i=0}^{N}(G_i - S_i)^2}}{\bar{G}_i} \tag{3}$$

$$Eff = 1 - \frac{\sum_{i=0}^{N}(G_i - S_i)^2}{\sum_{i=0}^{N}(G_i - \bar{G}_i)^2} \tag{4}$$

where $S_i$ denotes satellite rainfall values, $G_i$ represents gauge rainfall values and N the total number of data values in each satellite or gauge measurements.

## 5. Results and Discussion

### 5.1. Inter-Annual Comparisons of VRPs

The capabilities of the products to represent the annual rainfall patterns countrywide and in the prevailing agro-ecological zones of the country is provided in this section. Figure 2 represents the annual rainfall time series over the entire country and in the four zones. Countrywide, it is observed that all the VRPs showed a considerably good skill with r > 0.5. However, CHIRPS was observed to be the best with r = 0.74 and a minimal RMSE value of 0.04 (see Table 2). This was followed by GPCC and TRMM with RMSE values of 0.08 and 0.09, respectively. ARC2, on the other hand, showed a relatively high RMSE value of 0.22 (see Table 2). The good performance of CHIRPS in the region is mostly attributed to its blend of several input data including, in-situ precipitation observations, the Tropical Rainfall Measuring Mission (TRMM) 3B42, and many others. It leverages on the advantages of each input data; hence, it has a better performance than the rest of the products [19].

The poor performance of the ARC2 confirms previous studies [25,41] which have reported its inconsistencies particularly over the West African region. Regarding the agro-ecological zones (see Figure 2b and Table 2), the r values for the VRPs were in a decreasing order of 0.87, 0.86, 0.76, 0.74, 0.70, and 0.59 for CRU, CHIRPS, GPCC, ARC2, TRMM, and TAMSAT, respectively, in the Savannah zone. This indicates that gauge-only and merged satellite-based (CHIRPS) rainfall products tend to detect and measure annual rainfall pattern better in the Savannah zone than the other products. CHIRPS and TAMSAT showed relatively good skills with RMSE < 0.09, whereas TRMM and GPCC were relatively poor in the Transition zone. Although we may place more confidence in CRU and TAMSAT, CHIRPS revealed a better skill with correlation coefficient >0.90 in the Forest zone. Moreover, ARC2 was the worse in this zone with relatively large RMSE >0.32. A number of studies (e.g., [42,43]) have reported the weaknesses of satellite rainfall products over the coasts. Here, ARC2 performance with respect to gauge in the coasts of Ghana was observed to be worse

compared to the other products. In this zone, CRU was the best followed by CHIRPS with r values for both products >0.80. The performance of TRMM with respect to gauge was observed to drop in the Coastal zone as r value was <0.40 (see Table 2).

**Table 2.** Annual performance of VRPs with respect to gauge over the entire country and in the four zones for the period of 1998–2012.

| Dataset | Savannah | | Transition | | Forest | | Coast | | Countrywide | |
|---------|------|------|------|------|------|------|------|------|------|------|
| | r | RMSE | r | RMSE | r | RMSE | r | RMSE | r | RMSE |
| CMAP | – | – | – | – | – | – | – | – | 0.742 | 0.086 |
| GPCP | – | – | – | – | – | – | – | – | 0.784 | 0.063 |
| GPCC | 0.763 | 0.066 | 0.376 | 0.121 | 0.583 | 0.251 | 0.681 | 0.139 | 0.600 | 0.084 |
| ARC2 | 0.738 | 0.117 | 0.682 | 0.205 | 0.542 | 0.324 | 0.531 | 0.558 | 0.759 | 0.221 |
| CRU | 0.872 | 0.055 | 0.594 | 0.093 | 0.788 | 0.118 | 0.850 | 0.124 | 0.884 | 0.050 |
| TAMSAT | 0.591 | 0.080 | 0.637 | 0.082 | 0.747 | 0.092 | 0.535 | 0.138 | 0.713 | 0.075 |
| TRMM | 0.697 | 0.093 | 0.393 | 0.104 | 0.539 | 0.103 | 0.313 | 0.213 | 0.509 | 0.089 |
| CHIRPS | 0.858 | 0.059 | 0.659 | 0.084 | 0.905 | 0.068 | 0.802 | 0.070 | 0.928 | 0.042 |

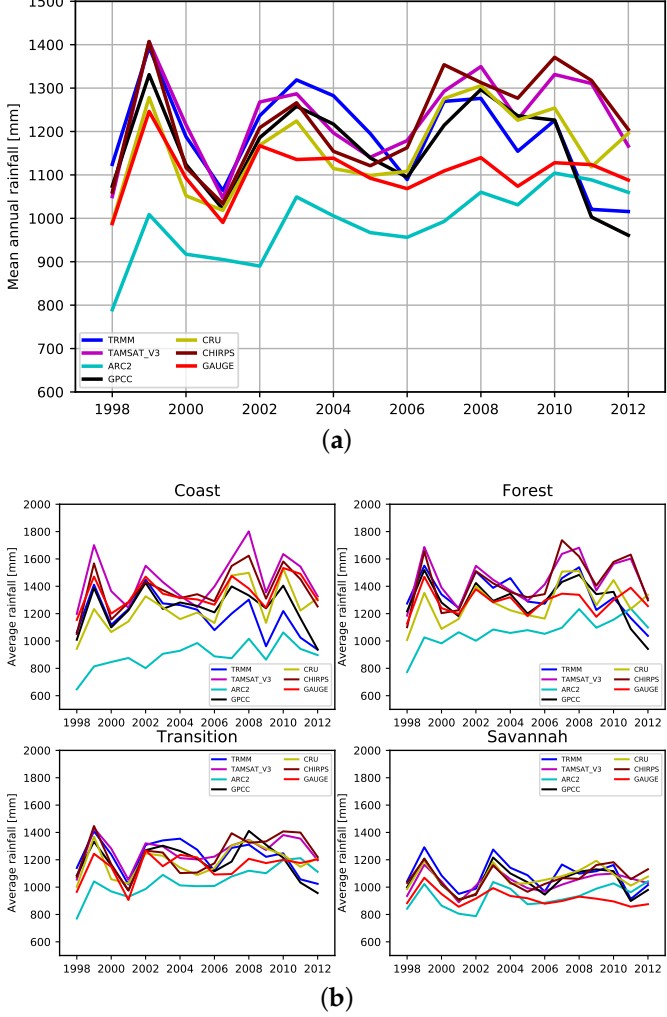

**Figure 2.** Annual rainfall time series for a period of 1998-2012 over the entire country (**a**) and in the four agro-ecological zones (**b**); GPCC (black), ARC2 (cyan), TRMM (blue), TAMSAT (magenta), CRU (yellow), CHIRPS (brown), and gauge (red).

The annual rainfall departures between the VRPs and observations countrywide and in the four agro-ecological zones are shown in Figure 3. This gives a very illustrative picture, showing a clear indication of where these VRPs are performing better or worse in capturing the wets and drys with respect to the gauge in the region. It is observed that all VRPs seem to have generally overestimated gauge. ARC2 was observed to be very dry with respect to gauge with bias >100 mm/year throughout the entire country. In addition, ARC2 was observed to underestimate gauge whilst the other VRPs overestimated approximately by less than 200 mm/year in the Savannah zone. Overall, all except ARC2 were observed to overestimate gauge in the Transition, Forest, and Coastal zones of the country (Figure 3b). The underestimations observed in ARC2 may be attributed to the unavailability of daily GTS gauge reports in real time, and perhaps deficiencies in the satellite estimate associated with precipitation processes over Coastal and orographic areas [43].

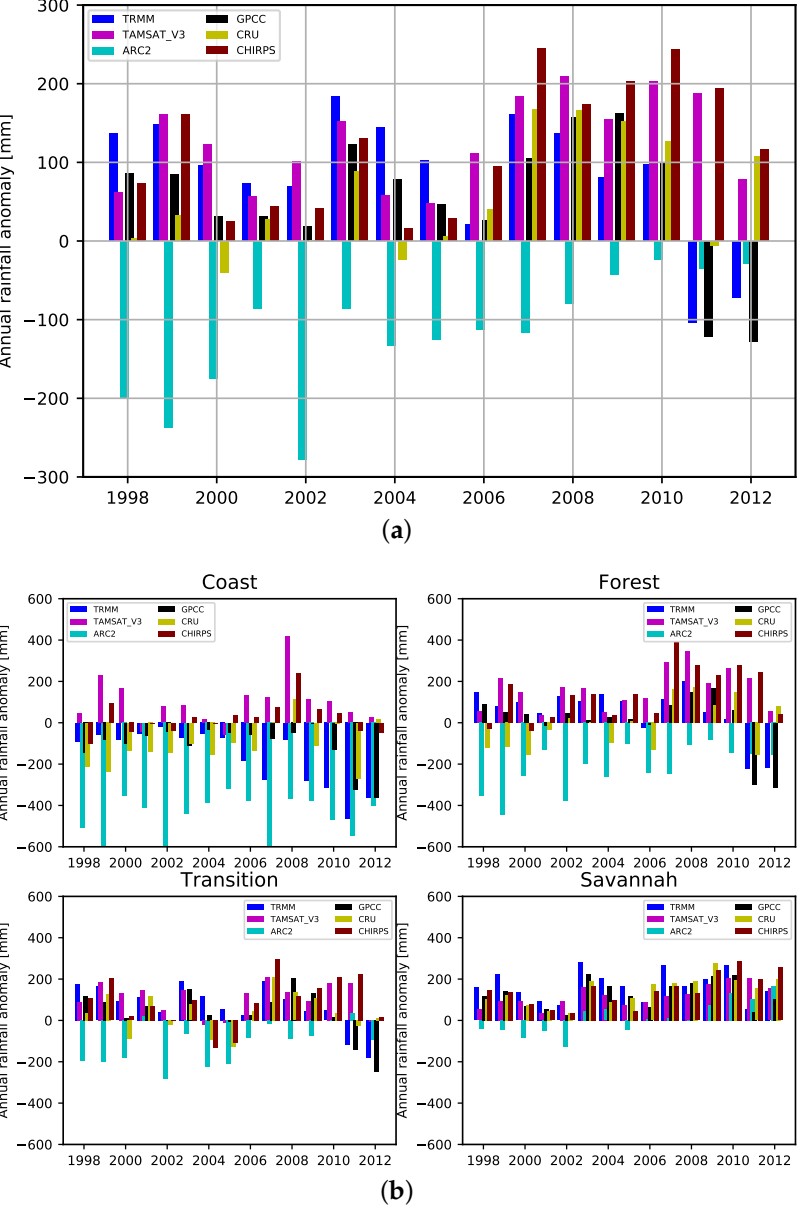

**Figure 3.** Annual sensor-based deviation from gauge records for the period of 1998-2012 over the entire country (**a**) and in the four agro-ecological zones (**b**); GPCC (black), ARC2 (cyan), TRMM (blue), TAMSAT (magenta), CRU (yellow), CHIRPS (brown), and gauge (red).

## 5.2. Inter-Seasonal Comparisons of VRPs

The results for the inter-seasonal performance of the VRPs over the entire country and in the four zones are presented here. Figure 4 shows a countrywide bias between VRPs and gauge for the study period. An overestimation was seen in the warm rain season (May–September) for all VRPs but ARC2. The periods of 2010–2012, however, showed a dry bias for TRMM and GPCC. Although it is an expectation that the gauge only data sets (CRU and GPCC) agree well with gauge observations, there appear to be some discrepancies which coincide with a sharp drop-off in gauge coverage beyond 2005 (see Figure 1d). These reflect likely calibration errors arising from the fewer gauges [44].

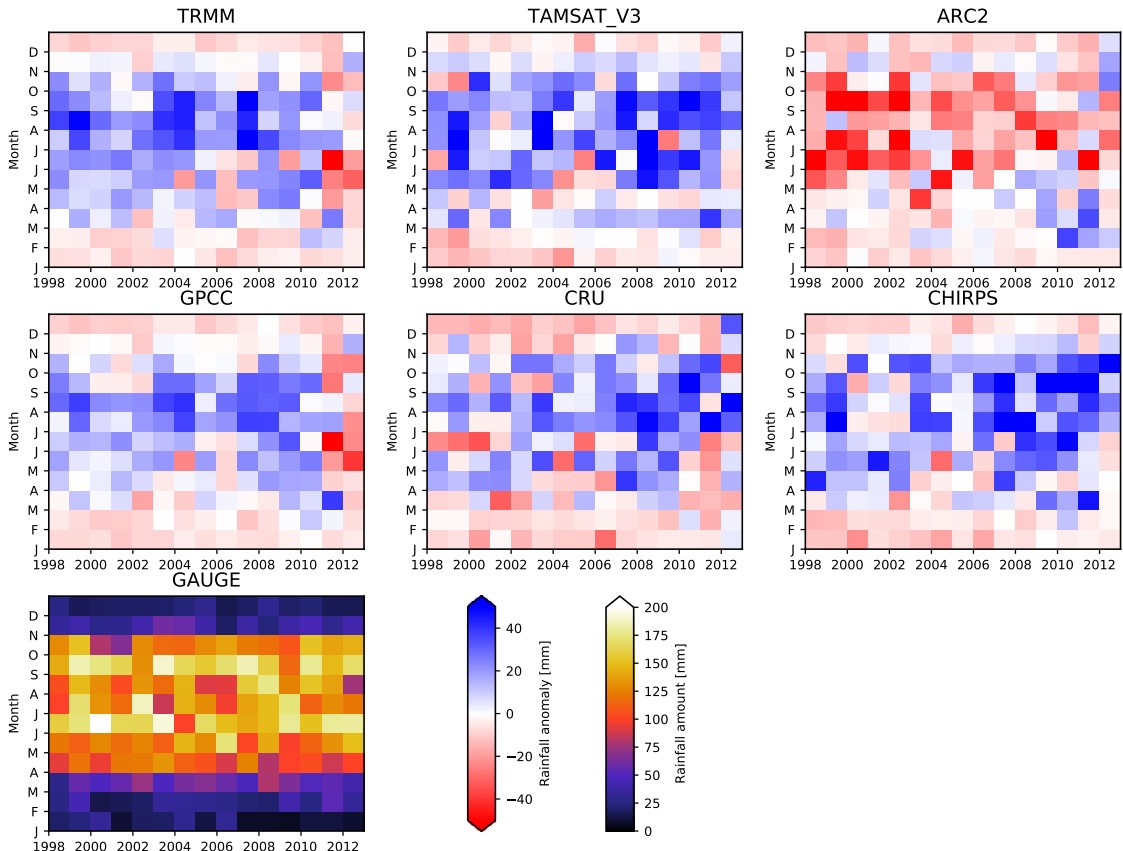

**Figure 4.** Countrywide bias between validated rainfall products and gauge for the period of study.

Figure 5 shows the spatial pattern of annual rainfall bias between the VRPs and gauge over the country. We observe a good representation of the rainfall patterns by all VRPs over the entire country, although there were instances of under/over estimations similar to findings in Pfeifroth et al. [45] over Tropical Africa. The biases are likely associated with shortcomings in the satellite retrieval algorithms. For instance, PM algorithms confuse warm orographic rains and very cold surfaces with other precipitation forms while IR algorithms have challenges in capturing warm orographic rains [46,47]. CMAP and GPCP were wetter than gauge in some parts of southern Ghana during January–May. In addition, TRMM, TAMSAT, GPCC, CHIRPS, and CRU showed positive (wet) bias during the July–September period in the far North of the country (see Figure 5). Evidence from Amekudzi et al. [27] and Atiah et al. [5] suggest sparsity of gauge network in this area, which could reduce the accuracy of inter-comparison. On the contrary, ARC2 largely underestimated gauge throughout the entire country (Figure 5). This dry bias of ARC2 in the country is attributable to lack of Global Telecommunication System (GTS) gauge records, coupled with the evidence that the temperature threshold used for ARC2 are too low for the West African region [25,44]. The wet biases seen in the northern part of the country by all VRPs and likely linked with convective clouds formed under dry conditions with cloud bases considerably higher than those formed under a moist

environment [48]. Thus, there is increased evaporation of precipitating rains (virga) which makes VRPs liable to overestimate the true rainfall as captured by gauge. Additionally, all VRPs were able to capture the onsets, cessations, and the spells (wet and dry) over the country. This particularly agrees with findings reported about some satellite rainfall products in Amekudzi et al. [27] and Usman et al. [26] over Ghana and Nigeria, respectively.

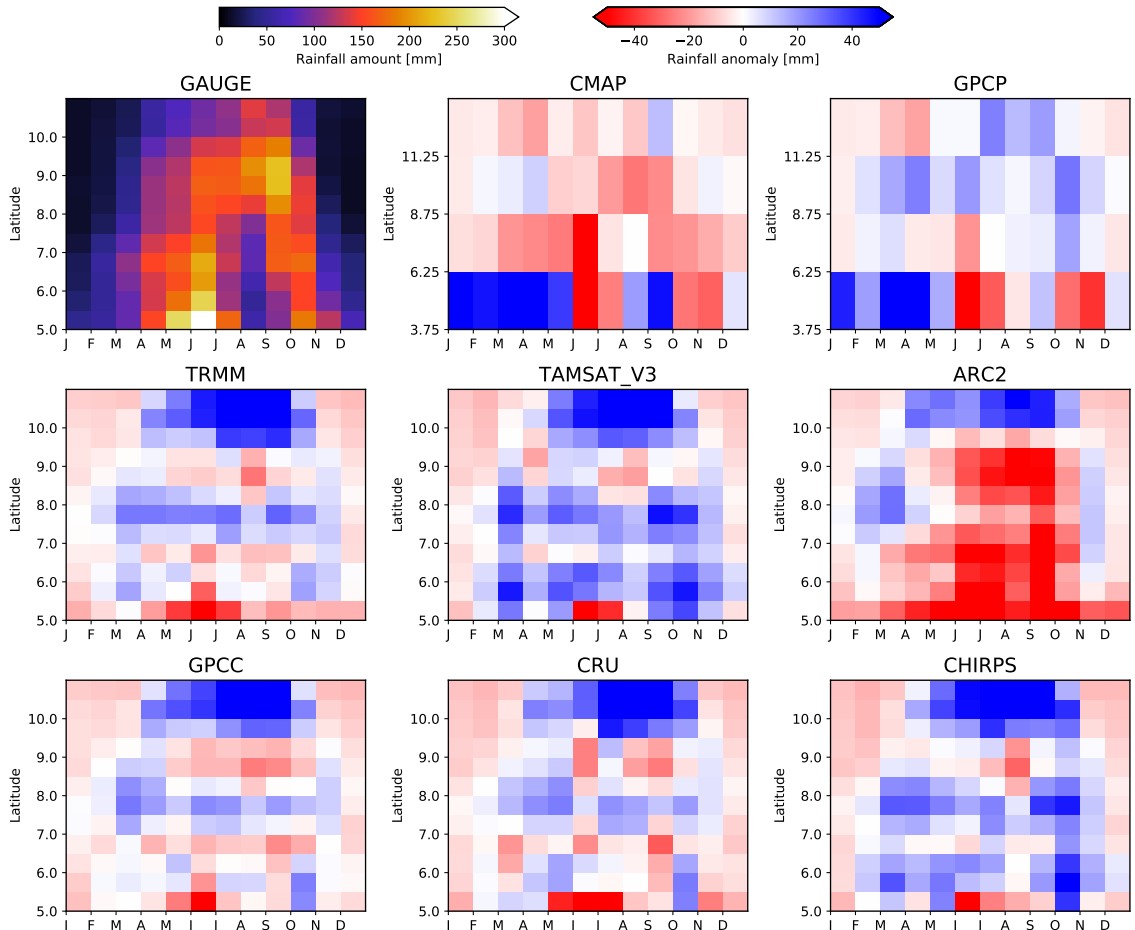

**Figure 5.** Spatial pattern of the monthly rainfall anomalies between the VRPs and observation over Ghana for the period of study.

Figure 6 shows the scatter plots of the VRPs for the period of study. This assesses the consistency of monthly rainfall between the VRPs and gauge. Generally, all VRPs performed very well with r > 0.90. CHIRPS was observed to be the most efficient followed by CRU, TRMM, and TAMSAT with eff values 0.95, 0.94, 0.93, and 0.92, respectively. ARC2, CMAP, and GPCP were the least performing VRPs, which is evident from the large RMSE and low eff values (see Table 3). Figure 7 represents the mean seasonal rainfall patterns captured by VRPs and the gauge in the four zones. The uni-modal rainfall pattern was well-captured by the VRPs in the Savannah zone. The peak rainfall amount (approximately 180 mm) in August was slightly overestimated by all VRPs. The bi-modal rainfall pattern in the Transition, Forest, and Coastal zones (with peak rainfall in June and September/October) were well-captured, which is consistent with findings of Aryee et al. [2]. Furthermore, peak rainfall amount in September was slightly overestimated by TRMM, CHIRPS, and TAMSAT, and underestimated by CRU, GPCC, and ARC2 in the Transition zone. Again, peak rainfall in October was slightly underestimated by ARC2 and overestimated by the other VRPs in the Forest zone.

**Table 3.** Monthly performance of VRPs with respect to gauge over the entire country for the period of 1998–2012.

| Satellite Product | RMSE | Eff | r | Bias |
|---|---|---|---|---|
| CMAP | 0.276 | 0.769 | 0.928 | 0.938 |
| GPCP | 0.217 | 0.865 | 0.953 | 0.960 |
| GPCC V5 | 0.167 | 0.936 | 0.970 | 0.967 |
| ARC2 | 0.359 | 0.643 | 0.949 | 0.824 |
| CRU | 0.169 | 0.938 | 0.970 | 0.969 |
| TAMSAT | 0.179 | 0.922 | 0.963 | 1.053 |
| TRMM | 0.171 | 0.933 | 0.966 | 0.992 |
| CHIRPS | 0.152 | 0.948 | 0.975 | 1.021 |

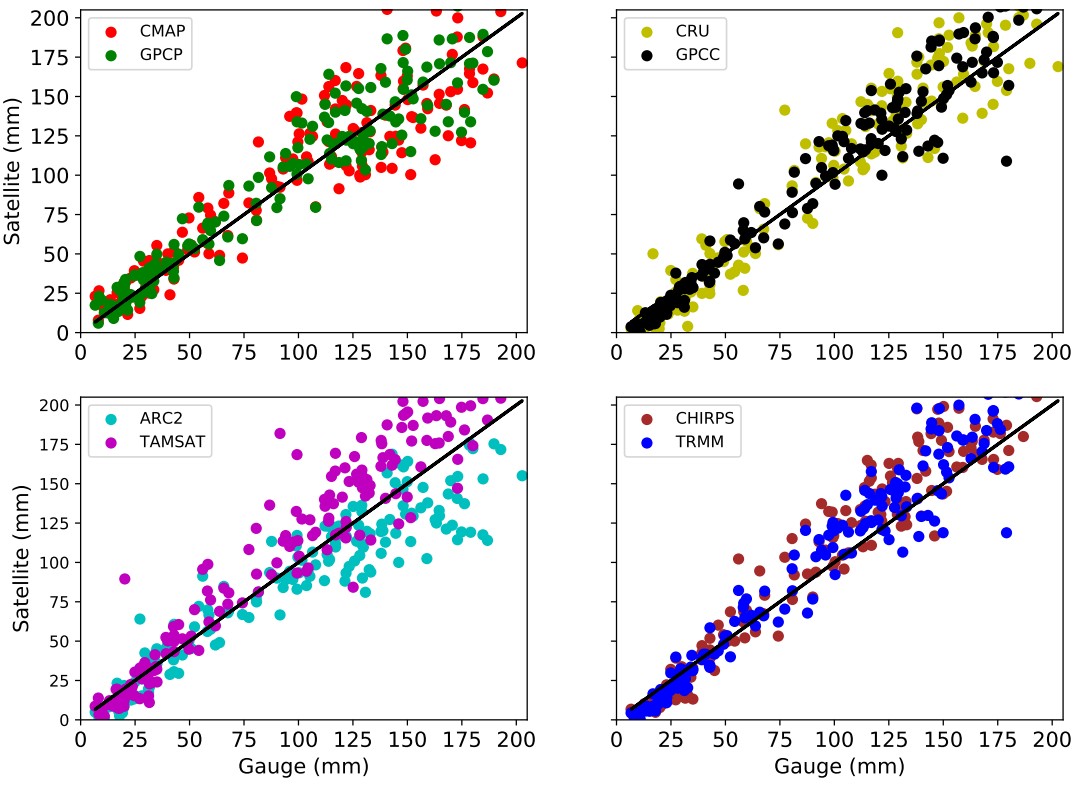

**Figure 6.** Scattergrams of monthly rainfall records for VRPs with respect to gauge over the entire country for the period of 1998–2012 ; GPCC (black), ARC2 (cyan), TRMM (blue), (red), CMAP (green), TAMSAT (magenta), CHIRPS (brown), and CRU (yellow).

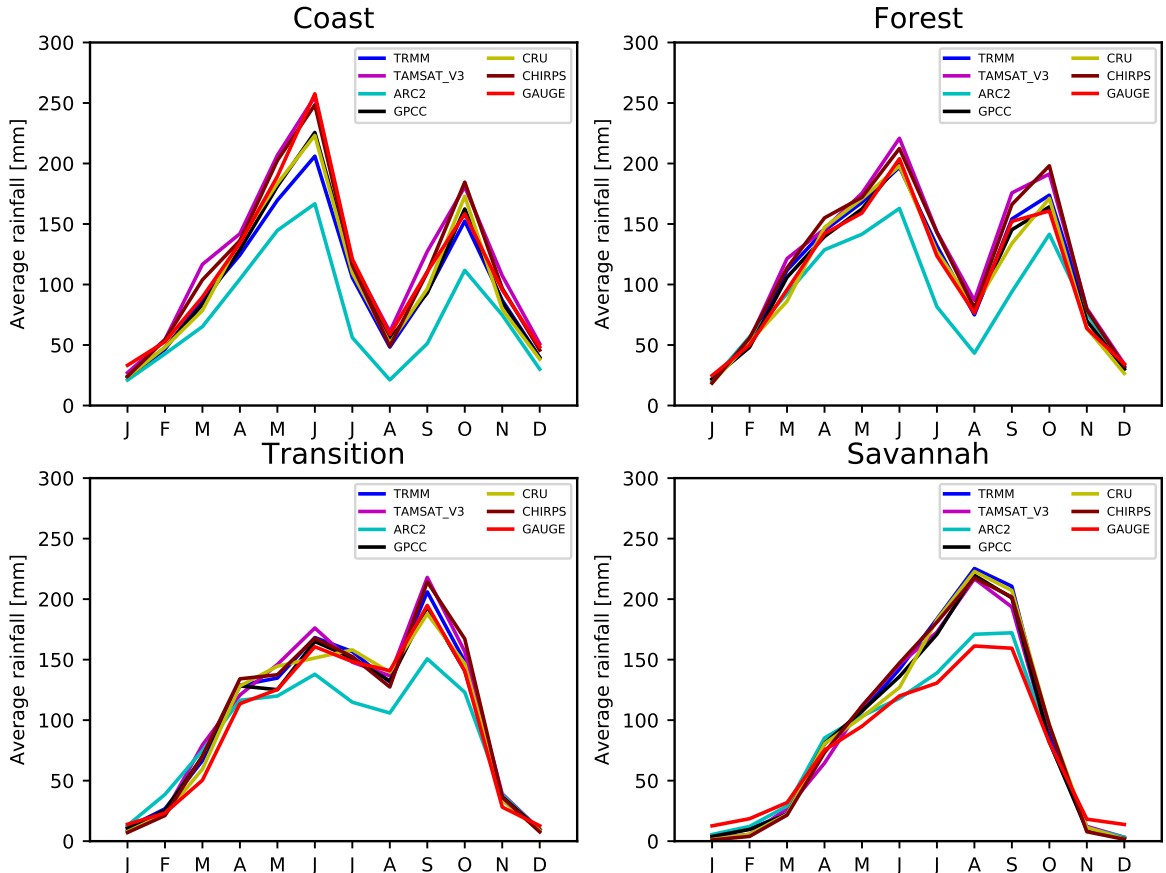

**Figure 7.** Inter-zonal monthly rainfall patterns for Gauge (red), TRMM (blue), TAMSAT (magenta), ARC2 (cyan), GPCC (black), CRU (yellow), and CHIRPS (brown) rainfall products.

*5.3. Statistical Evaluation of VRPs*

To investigate the performance of the VRPs statistically, Taylor diagrams, which provide a summary of the relative skill of the VRPs with respect to gauge were constructed countrywide and in the four agro-ecological zones. The yellow solid and dashed lines represent the observed (GMet v1.0) and the minimum Centered Root Mean Square Error (CRMSE) of the gauge respectively, and the grey line is the CRMSE.

Figure 8 shows the monthly rainfall cycle of the VRPs over the entire country. It is observed that all satellite rainfall products showed very good agreement with gauge over the entire country with r values >0.90. CHIRPS was the best among the VRPs in the entire country with the least RMSE value of approximately 15 mm and relatively higher r values. CRU and GPCC showed similar strength of agreement with gauge with r > 0.93 and RMSE <20 mm over the entire country. Furthermore, all rainfall products except TAMSAT and CHIRPS slightly underestimate gauge throughout the entire country. The least performing VRP was ARC2 with relatively larger RMSE and bias values of 24 mm and 18 mm, respectively.

Figure 9 shows the performance of the various products after removal of seasonality from their respective rainfall data in comparison with the gauge in the entire country. We found that the seasonal performance of the VRPs were relatively better than after removal of seasonality as r values were less than 0.80. CHIRPS was observed to be the best followed by CRU, GPCC, and TRMM, while the remaining VRPs had r values <0.65. This thereby suggests that the VRPs sometimes captured rainfall relative to the gauge on average, however, the amount recorded by gauge was either overestimated or underestimated. Another possible cause could be attributable to the interpolation error in the gauge data resulting from fewer gauges in a grid cell, which could produce inaccurate estimates of rainfall in the gauge data. Again, with regard to the VRPs, since the satellite estimates

are adjusted by the gauge measurement, this implies that the satellite estimates may sometimes be closer to the truth than the blended gauge and satellite product.

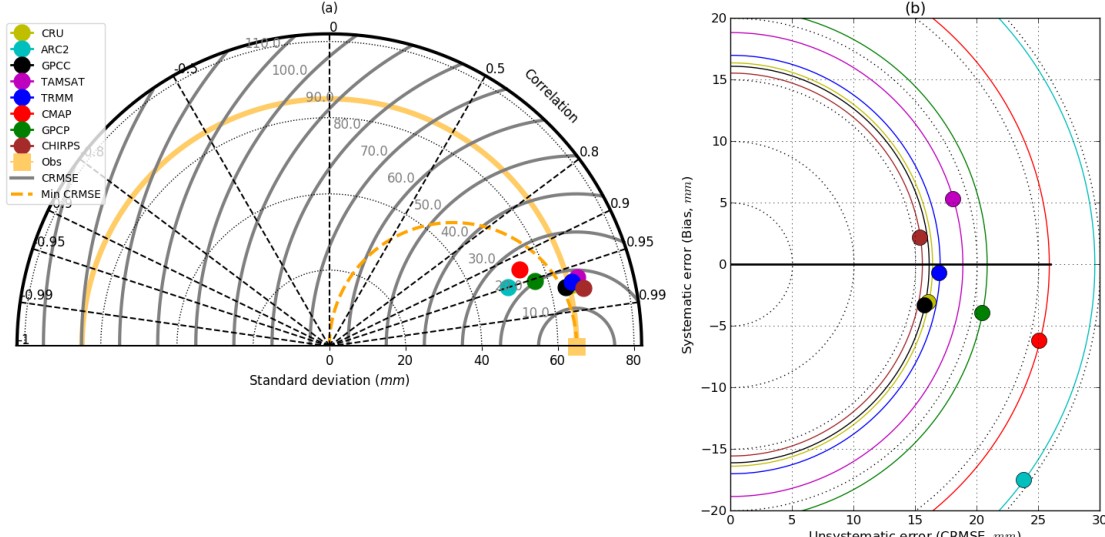

**Figure 8.** Taylor (**a**) and error (**b**) diagrams on monthly data for the period of 1998–2012 showing the performance of the GPCC (black), ARC2 (cyan), TRMM (blue), (red), CMAP (green), TAMSAT (magenta), CRU (yellow), and CHIRPS (brown) for the entire country.

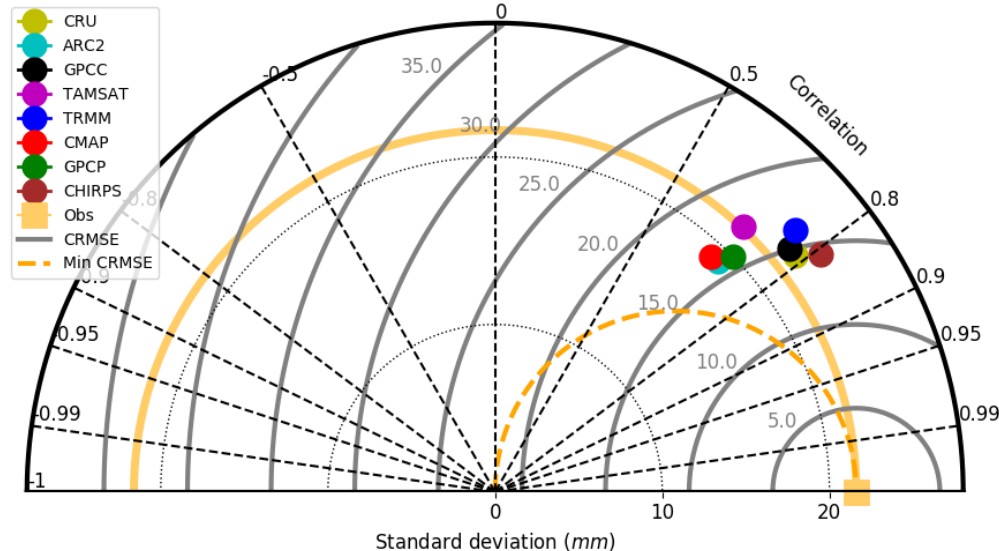

**Figure 9.** Monthly anomalies showing the performance of CRU (yellow), TAMSAT (magenta), TRMM (blue), GPCC (black), CHIRPS (brown), and ARC2 (cyan) with respect to gauge (red).

Figure 10 depicts the inter-zonal Taylor diagrams whilst Figure 11 is the inter-zonal error diagrams showing the seasonal performance of the rainfall products in comparison with the gauge. All VRPs except ARC2 were observed to have similar strength of performance with very good correlation coefficients (r > 0.95) in the Savannah zone. There was a consistent agreement by all rainfall products with gauge except ARC2 which showed a slight underestimation in the Savannah zone. In addition, all products were observed to perform well in the Transition zone with r > 0.80; however, ARC2 underestimated gauge rainfall values. Again, all VRPs were observed to have good agreement with gauge (r > 0.80) and CHIRPS was found to have the least bias and RMSE values, whereas ARC2 recorded relatively large RMSE value. In general, rainfall values were underestimated by

all except CHIRPS and TAMSAT in the Forest zone. In the Coastal zone, all except TAMSAT and CHIRPS underestimate rainfall values with bias consistently less than 40 mm. A possible source of underestimation in TAMSAT is that the product is geared toward drought monitoring and so, accurately representing low rainfall amounts is given priority as the median rainfall is chosen, as it is insensitive to the high rainfall event, while being more representative of typical, lower rainfall amounts [44,48]. Another possible source of underestimation is that no gauge corrections have been applied to the gauge records used in the TAMSAT calibrations [44,49]. The GPCC, CRU, and CHIRPS were observed to have relatively small bias and RMSE values, whereas ARC2 had the largest RMSE and bias values. The general performance of all satellite rainfall products in all zones was much closer except for ARC2 in the Savannah, Transition, and Forest zones than their performance in the Coastal zone.

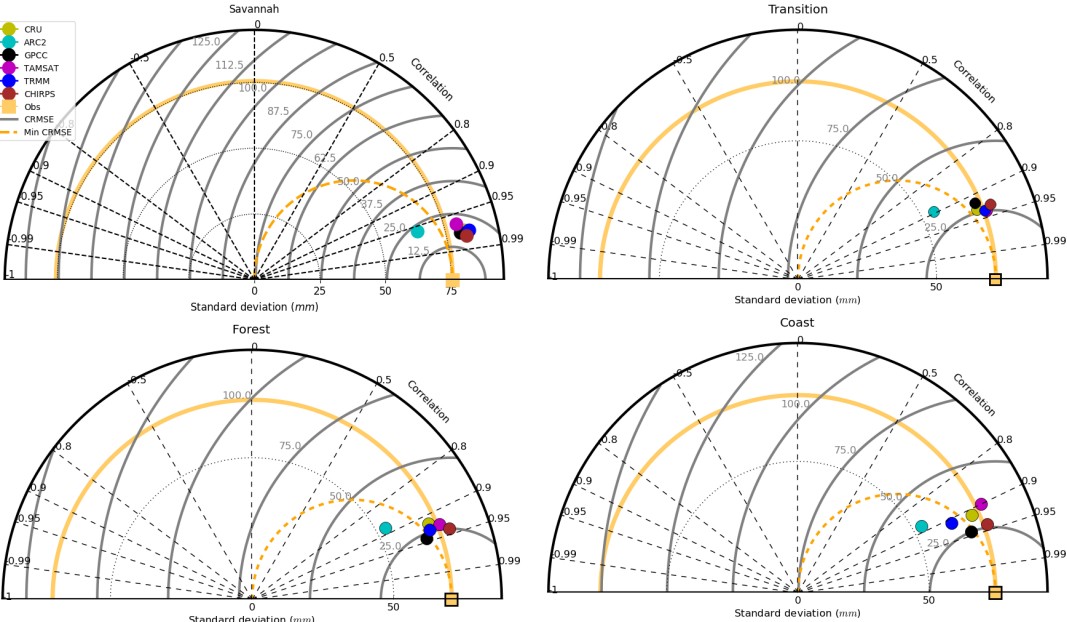

**Figure 10.** Inter-zonal Taylor diagrams showing the performance of CRU (yellow), TAMSAT (magenta), TRMM (blue), GPCC (black), CHIRPS (brown), and ARC2 (cyan) with respect to Gauge.

On average, all VRPs were observed to perform relatively poor in all four ecological zones after removal of seasonality from their respective datasets (see Figure 12). Specifically in the Savannah zone, CHIRPS showed a relatively better skill followed by CRU, GPCC, and TRMM; however, correlation coefficients were <0.80 for all products. In the Transition zone, the correlation coefficients for all products were less than 0.60. Comparatively, CHIRPS and GPCC showed a relatively similar and better skill in the Forest and Coastal zones with r ≥ 0.80, whereas the remaining VRPs had r values <0.80. A possible reason for the relatively poor performance of the VRPs compared to observation after removal of seasonality points to the fact that the VRPs were mainly able to capture the seasonal rainfall pattern in the region; however, the quantum or amount of rainfall is not well captured. This indicates that the wets and drys in the data are poorly represented by them, resulting in the over/under estimations observed. This could also be attributed to satellite data measuring algorithms. In addition, PM algorithms have a tendency of confusing warm orographic rains and very cold surfaces with precipitation, while IR algorithms have challenges with warm orographic rains. With regards to the agro-ecological zones, one reason may be due to different rain patterns or processes in the different zones, which means using a single satellite algorithm or calibration may not be sufficient for use at different locations.

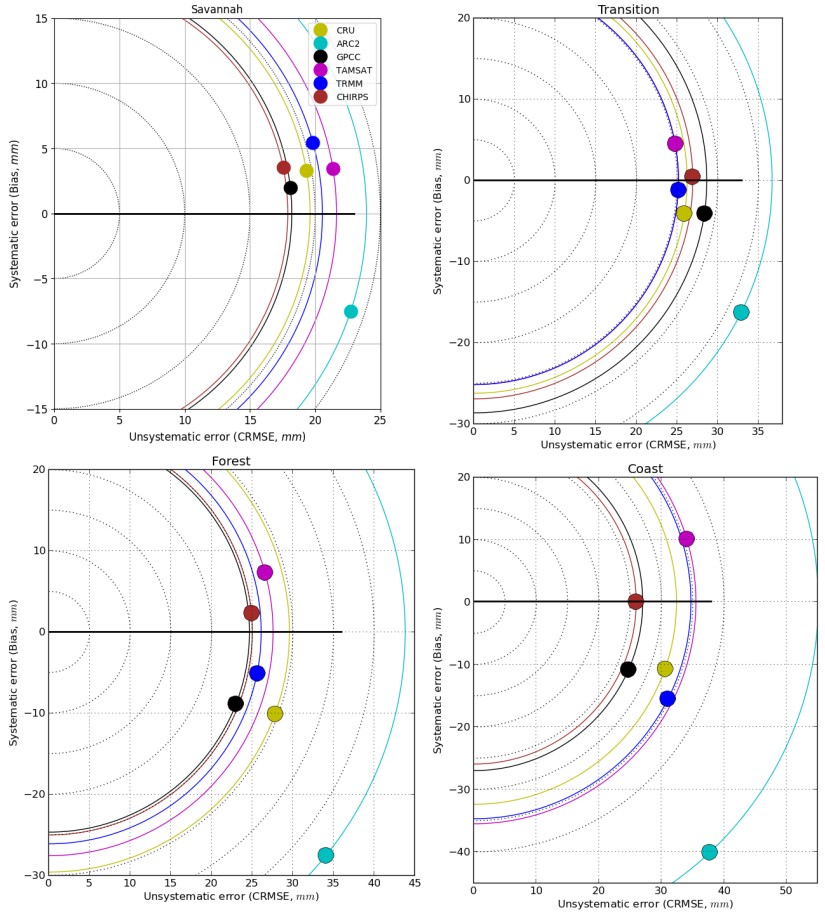

**Figure 11.** Inter-zonal error diagrams showing the performance of CRU (yellow), TAMSAT (magenta), TRMM (blue), GPCC (black), CHIRPS (brown), and ARC2 (cyan) with respect to Gauge.

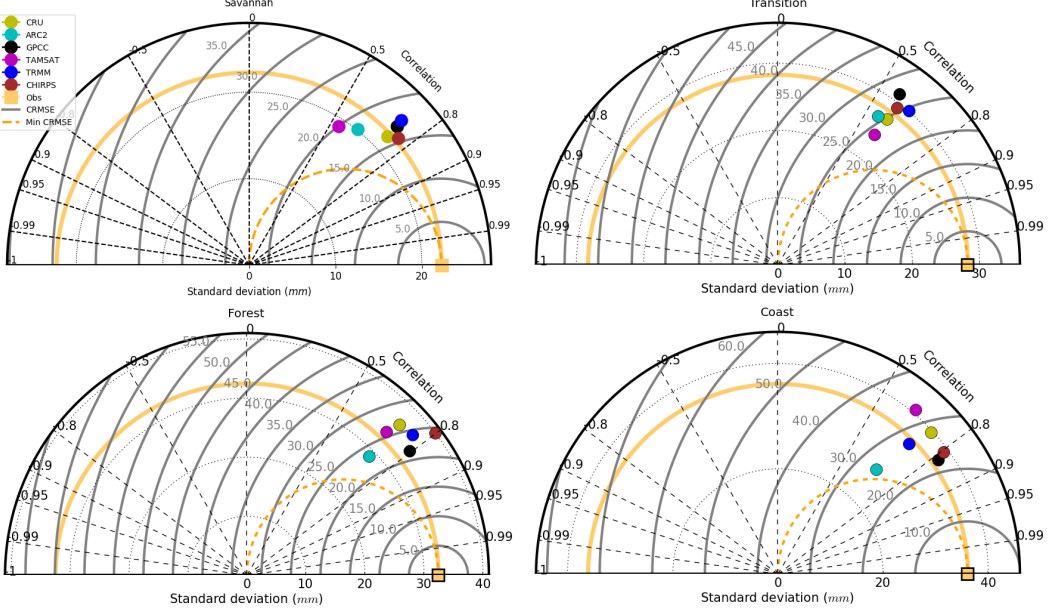

**Figure 12.** Inter-zonal Taylor diagrams showing the performance of the GPCC (black), ARC2 (cyan), TRMM (blue), TAMSAT (magenta), CRU (yellow) and CHIRPS (brown) after removal of seasonality for the period of 1998–2012.

## 5.4. Pixel–Pixel Evaluation of VRPs

In this section, the rainfall products were assessed taking into consideration the number of gauge stations in a grid cell for the four agro-ecological zones. This analysis proves crucial as it would increase our understanding on the minimum number of stations necessary for a meaningful satellite validation in a data-sparse region, a study which is limited in literature. For this analysis, the number of point stations in a given grid cell was used and thus no unique selection criteria were employed. Therefore, the Forest zone grids with 1, 2, 3, 4, and 5 stations were used, whereas grids with 1, 2, and 3 stations were applied for the Transition zone. On the other hand, grids with 1, 2, and 5 stations were analyzed in the Coastal zone. In addition, grid cells with 1, 2, and 4 gauge stations were used for this analysis in the Savannah zone. The arithmetic mean for each grid cell was then calculated for the raw gauge data, which was used as the reference in place of the gridded gauge data to avoid any possible interpolation biases. Thereafter, the VRP data for the corresponding grids of the gauge data were then extracted and Taylor diagrams constructed for these changing number of grid stations with respect to raw gauge data. On average, the results showed good agreement between the rainfall products and gauge data for all five grids with r values >0.60 in the Forest zone (see Figure 13). This indicates that the number of gauges per grid in this zone tends to have no/little impact on the performance of the VRPs. Similarly, in the Transition zone, good conformity was seen between the gauge and the VRPs (r > 0.50). CHIRPS and TRMM are noted to have variations that are similar to observations and are perceived to be less impacted by changing gauge numbers as compared to the other products in this zone (see Figure 14). In general, the results show that most of the products have good agreement (>0.60) with respect to gauge compared to grids with lower gauge numbers in the Coastal zone (see Figure 15). In addition, the grid cell with five point stations had the least RMSE values compared with the grid cells of a lower number of point stations. The Savannah zone reveals a similar vulnerability of the region to changing gauge numbers in a grid cell as was in the Coastal zone, thus, the VRPs tend to have better agreement with the gauge and significantly reduced RMS errors in the grid cell with four stations, whereas the RMS errors are larger in the remaining grids (see Figure 16).

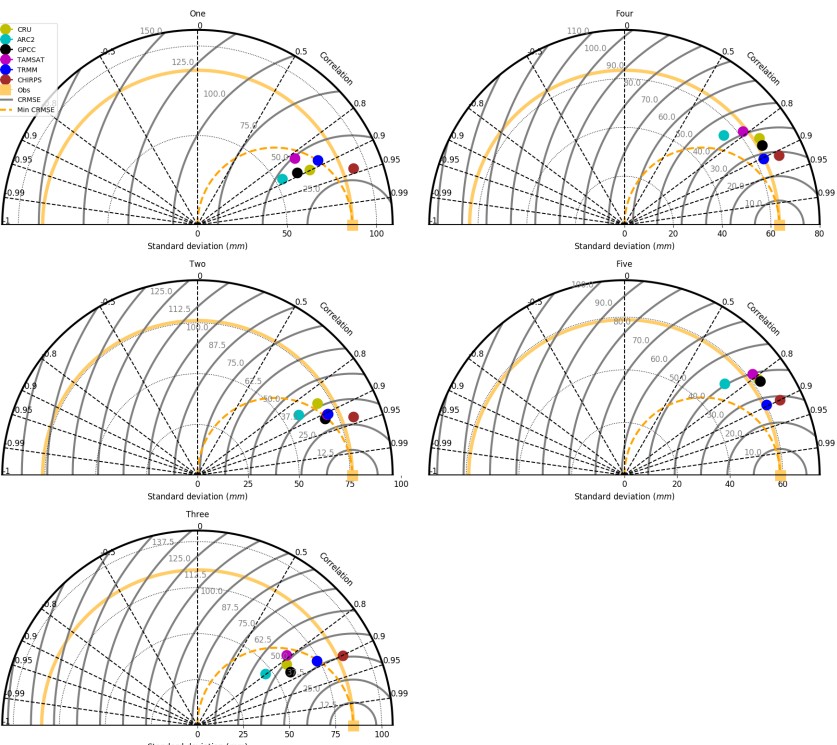

**Figure 13.** Taylor diagram showing the performance of the VRPs with changing grids and number of stations in the Forest zone.

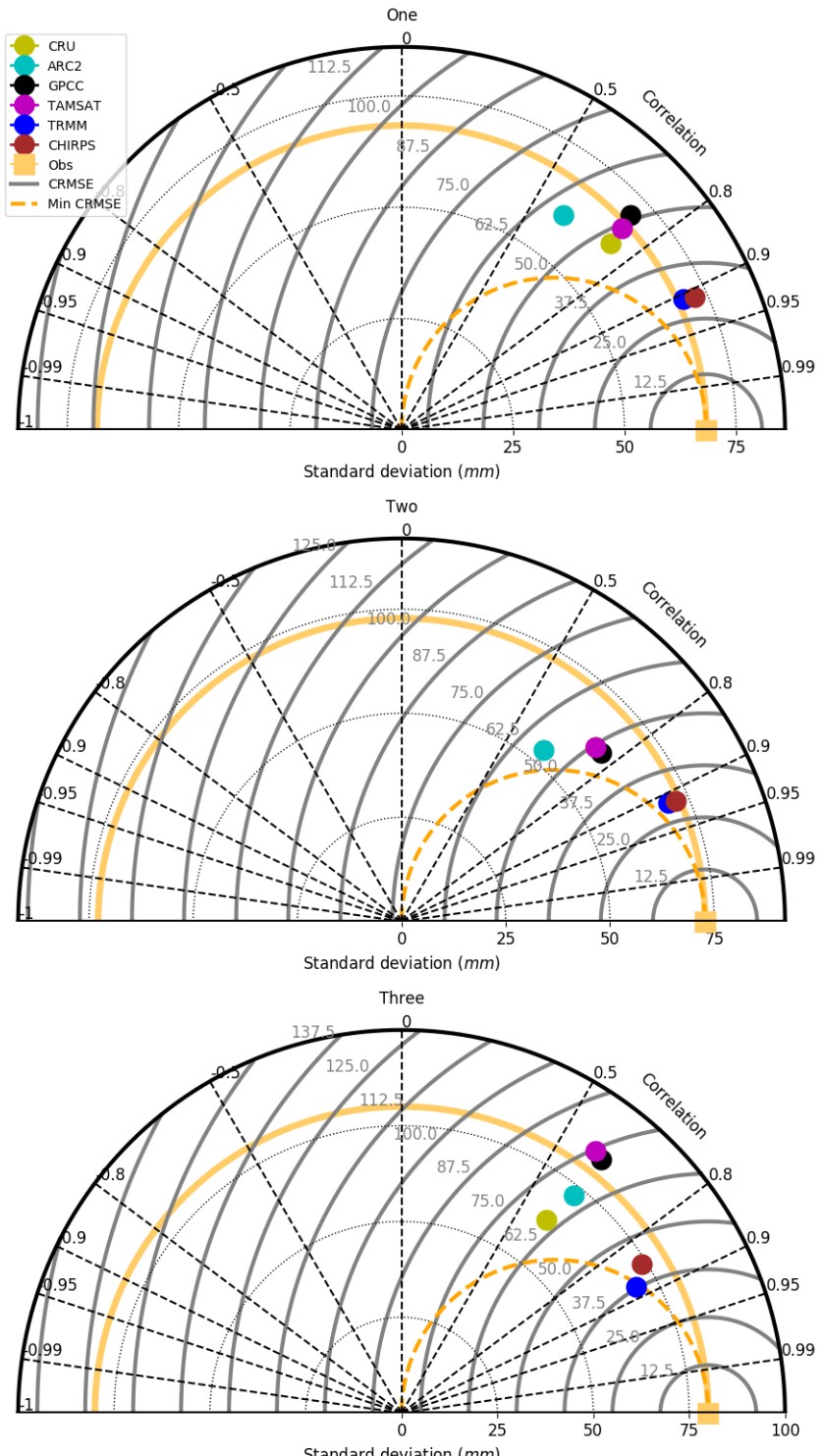

**Figure 14.** Taylor diagram showing the performance of the VRPs with changing grids and number of stations in the Transition zone.

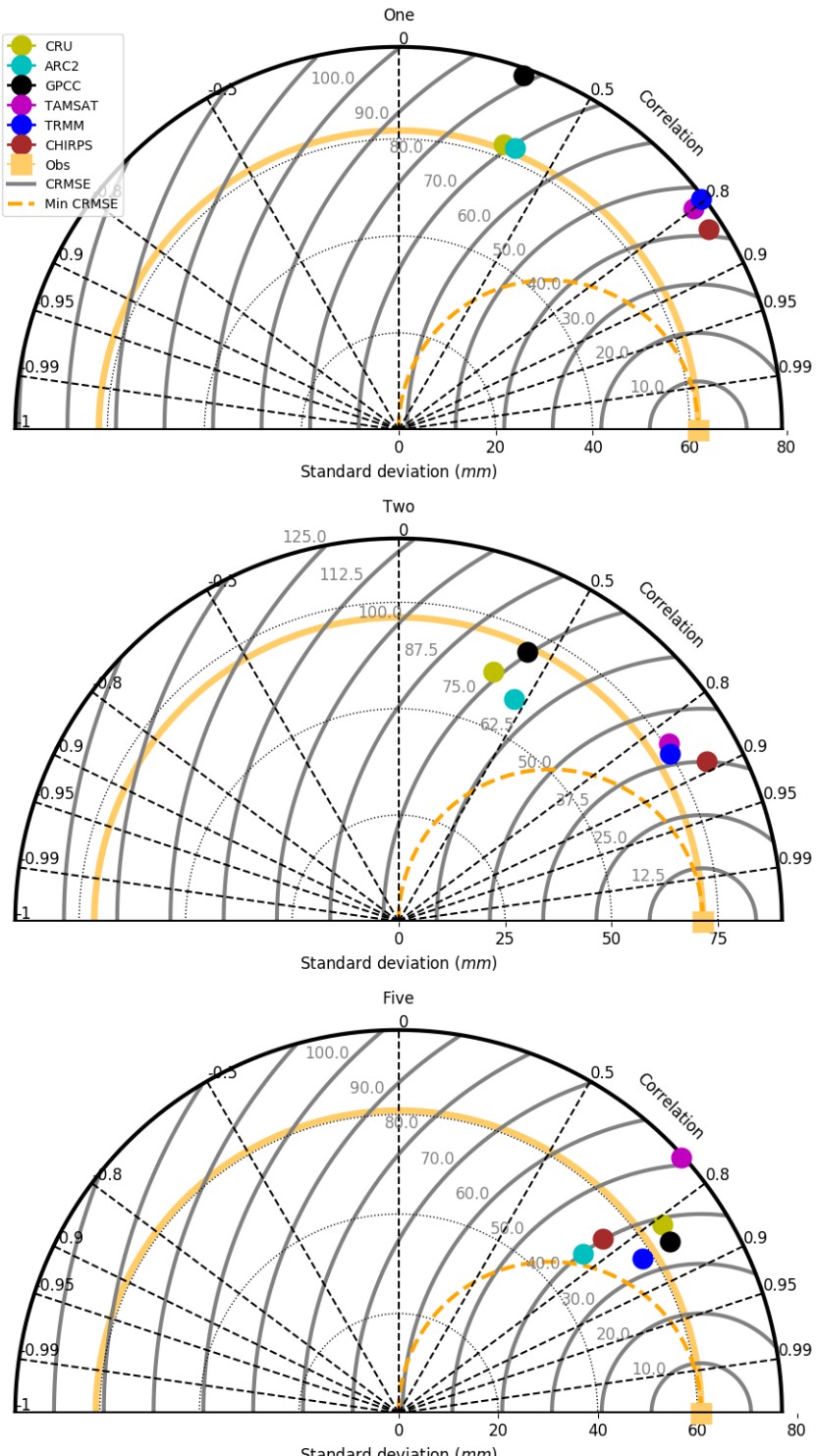

**Figure 15.** Taylor diagram showing the performance of the VRPs with changing grids and number of stations in the Coastal zone.

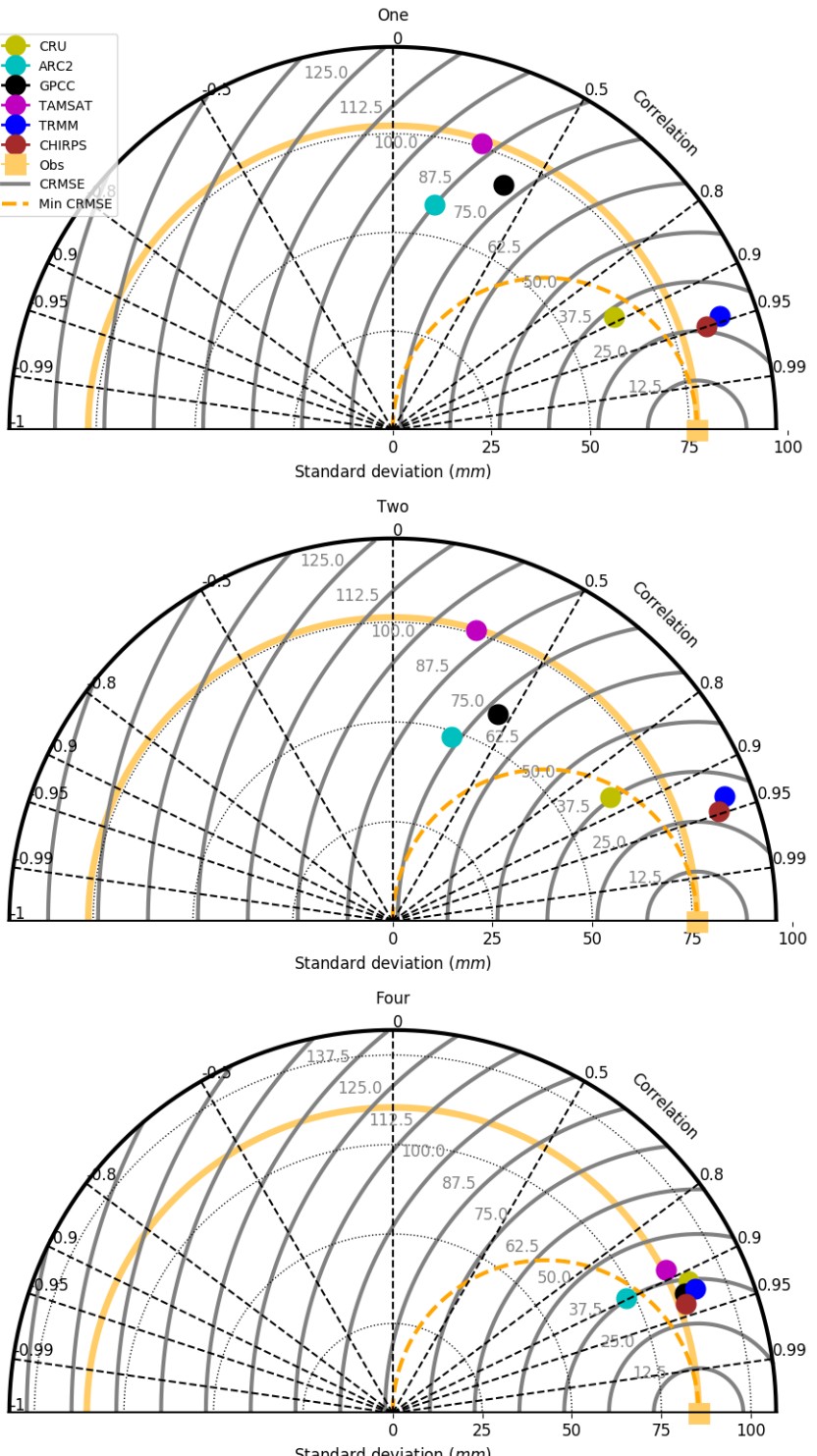

**Figure 16.** Taylor diagram showing the performance of the VRPs with changing grids and number of stations in the Savannah zone.

## 6. Conclusions

Satellite-based rainfall products are revealed through various studies to present crucial information on the rainfall variability in data-sparse West Africa. Nonetheless, satellite products are associated with many error types; therefore, assessing their performance is essential to boosting our confidence in their applications. In this study, a detailed validation of 2 gauge-only and 6 satellite-based rainfall products have been carried out over Ghana. The current study forms part of the Dynamic Aerosol Chemistry-Cloud-Interaction in West Africa (DACCIWA) project, Work Package (WP6), which aims to quantify the uncertainties in satellite rainfall estimates over West Africa. In addition, the study contributes to sub-regional validation studies that are limited in previous literature. In addition, the study assesses the reliability of the GMet v1.0 rainfall product. The performance of validated rainfall products in comparison with gauge was assessed over the country and in the four agro-ecological zones of the country. VRPs were assessed on seasonal and annual timescales using a suite of statistical techniques. The results revealed the abilities of VRPs to capture rainfall patterns in the four agro-ecological zones (Forest, Transition, Coastal, and Savannah). The onsets, cessations, and spells (wet and dry) over the entire country were also captured amidst few over/under estimations. Our results reveal the performance of the satellite-based rainfall products to be dependent on scale and location. The seasonal performance of all VRPs was relatively better than their annual performance. All VRPs except TAMSAT and CHIRPS were observed to underestimate gauge records in the country. In addition, the products showed relatively good skill in capturing the seasonal pattern of rainfall. However, removal of seasonality reduces the performance of the VRPs. Generally, CHIRPS showed better skill than other rainfall products; however, ARC2 largely under-performed. The results from point-pixel validation show the dependence of VRP performance on the number of point stations in a grid cell. This is, however, found to hold in the Coastal and Savannah zones of the country, whereas the Forest and Transition are less impacted by the number of gauge stations. Thus, for meaningful satellite validation, the minimum number of point stations in a grid cell required in the Savannah and Coastal zones should be $\geq 3$. For climate impact studies based on fine temporal resolution rainfall products, CHIRPS, TRMM, and TAMSAT would be great substitutes in the absence of gauge records. CHIRPS, CRU, and TAMSAT could also serve as complements to gauge for annual rainfall studies. GPCP could complement gauge records for studies involving coarse spatial rainfall resolution. Within the DACCIWA project, seventeen portable optical rain gauges with a minute temporal resolution have been deployed over the Ashanti Region of Ghana. Maintaining these gauge stations will serve as a valuable data source for high impact rainfall dynamics and validation studies.

**Author Contributions:** Conceptualization, W.A.A.; Methodology, W.A.A.; Software, W.A.A. and J.N.A.A.; Validation, W.A.A.; Visualization, W.A.A. and J.N.A.A.; Writing—original draft preparation, W.A.A.; Data curation, W.A.A. and J.N.A.A.; Supervision, L.K.A., S.K.D., and K.P.; Writing—Reviewing and Editing, J.N.A.A., W.A.A., L.K.A., S.K.D., and K.P. All authors have read and agreed to the published version of the manuscript.

**Funding:** The research leading to these results has received funding from the European Union 7th Framework Program (FP7/2007-2013) under Grant Agreement No. 603502 (EU project DACCIWA: Dynamics-aerosol-chemistry-cloud interactions in West Africa). The first author acknowledges support of the Organization of Women in Science for the Developing World (OWSD) and the Swedish International Development Cooperation Agency (Sida). The authors would like to acknowledge the support of the GCRF African SWIFT project.

**Acknowledgments:** We acknowledge Andreas Fink and Marlon Maranan of Karlsruhe Institute of Technology (KIT)-Germany and the two anonymous reviewers for in-depth scientific discussions.

**Conflicts of Interest:** The authors declare no conflict of interest.

## Abbreviations

The following abbreviations are used in this manuscript:

| | |
|---|---|
| GPCP | Global Precipitation Climatology Project |
| GPCC | Global Precipitation Climatology Center |
| TRMM | Tropical Rainfall Measuring Mission |
| CRU | Climate Research Unit |
| ARC2 | African Rainfall Climatology |
| TAMSAT | Tropical Applications of Meteorology Using Satellite Data and Ground-Based Observations |
| CHIRPS | Climate Hazards Group Infra-Red Precipitation with Station Data |
| CMAP | Climate Prediction Center (CPC) Merged Analysis of Precipitation |
| VRP | Validated Rainfall Product |

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
