# Peer review of "Validation of Satellite and Merged Rainfall Data over Ghana, West Africa"

_atmosphere, doi:10.3390/atmos11080859_

Round 1

Reviewer 1 Report

Review of the manuscript “Validation of Satellite and Merged Rainfall Data over Ghana, West Africa” by Atiah et al. This study is on the validation of rainfall products using GMet rain gauges over Ghana. The authors validated the products over the country, as well as for each region of the four ecological zones. The validation included annual, interannual, seasonal, latitudinal evaluations. The authors further investigated the impacts of the number of stations in the gauge grid on the validation result. While the manuscript is well written in terms of the English language, it overall lacks the scientific discussion of the results and the detailed explanation of the gauge datasets used in the analysis. Some descriptions even describes about different figures/tables from those identified in the text. Based on the comments summarised below, I would recommend a major revision in the current form. Major Comments: Throughout the paper, the manuscript compares the rainfall products against the gauge observation, discussing overestimations and underestimations, but significantly lacks the scientific explanations and interpretations on the reasons for those overestimation/underestimation - if they were attributed to satellite measurement or gauge observation or even to the nature of precipitation formation. The authors have not sufficiently presented the reliability of the gauge dataset used in this study. Clarification on how the gauge gridded data are retrieved from station observations should be indicated in Section 2.1. Given the high spatiotemporal variability of rainfall, the gauge data without an observation station are considered highly uncertain, which would affect the evaluation results presented. The number of gauges in each grid should also be presented (such as in Figure 1b). If the recommendation in the conclusion was to use the grids with 3 stations or more, why not conducted the overall analysis under this condition? Minor Comments: Section 2.1: Clarify the spatial resolution of the gauge gridded dataset and the temporal resolution of the original gauge data before merging to the monthly dataset. Line 161-165: Figure 2 does not show r and RMSE values. Line 165-166: The RMSE of ARC2 is 0.369 in Table 2. Line 170-184: Is this paragraph explaining the r and RMSE in Table 3 (not Figure 2)? If so, indicate at the beginning of the paragraph. Line 188-189: Most of the VRPs seems to overestimate compared to the gauge. Line 189-190: According to Figure 3a, there’s only a few months when ARC2’s bias was around 250 mm/year. Line 192: Most of the VRPs were overestimating precipitation in the Transition, Forest and Coastal zones. Line 202-209: Are these sentences explaining Figure 5 (not Figure 4)? Figure 2 and 4: There seems to be no detailed explanations/discussions on Figure 2 and Figure 4. Line 222-224: Figure 5 does not present yearly information (e.g. 2003-2007 and 2010-2012). Line 234-235: According to Figure 7, ARC2 also overestimated the peak rainfall in August in the Savannah Zone. Line 239-240: According to Figure 7, TRMM, CRU and GPCC seems to overestimate the peak rainfall in October in the Forest Zone. Line 245: What do you mean by seasonal performance? Did you select one season in Figure 8? If so why? Figure 8: Explain the yellow (solid and dashed) and gray lines in the main body, otherwise remove from the figure. Figure 9: The captions denotes “time series” but the figure is not showing time series. Line 253: Clarify how the seasonality was removed. Line 255, 274: What are the possible reasons for the VRPs performing worse after the removal of seasonality? Figure 2, 3, 7, 9-12: Discuss the scientific explanations/interpretations for the differences in VRPs among the agro-ecological zones. Taylor and error diagrams: Why are there two circles/squares for each VRPs shown in the diagram legends?

Author Response

Dear sir/madam,

Please kindly find attached our responses to your comments. We would also like to use this opportunity to thank you very much for your constructive contributions and concerns.

Reviewer 2 Report

Editorial Comments:

Please pay attention to paragraph indention.

Please make sure your paragraphs are not composed of two sentences. Otherwise, they look like floating ideas and your narrative breaks down.

General Comments:

Why images at the 4.4. Pixel-pixel Evaluation of VRPs go to the appendix if 4.4 is part of the evaluation process?

How did you compensate for the edge effects? As you know, there are no strict boundaries between ecological zones. Is it possible to quantify the impact of edge effect; or is it possible to enhance the methodology so that the edge effect is minimized?

Figure 1: Please either match the latitudes so that one can have a horizontal comparison (it’s shifted with some minutes). Or, if possible, these two figures. If you remove the grid points, you should have an easy-to-read map, and the reader will have a better understanding of the relationship between gauge distributions and ecological zones. Also, why do you include grid points on the sea (below 5degrees North)? They don’t go into the analysis, do they?

Figure 3: This figure is misleading. Figures should be self-explanatory. Mean annual rainfall on the Y-axis suggests there is negative rainfall. The minimum cannot be less than zero. If you are depicting the sensor-based deviation from measured gauges, labels on these axes should indicate that.

Figure 3. Do you know what makes GPCC throw off its measurements in Forest and Transition areas after 2010? As a follow-up question, is it then reliable to use GPCC after 2010 in your analysis?

In your text, you should mention the visualization/computation software you used.

In-line Comments:

  1. Please fully define CMRSE – centred root-mean-square error
  2. How did you perform “removal of seasonality”? There is no mention of time-series analysis in your work.

Overall

While I consider your work extremely useful and acknowledge why it is essential to cross-compare VRPs, I also think your methodology is straightforward and lacks an innovative aspect. As you have indicated, it is crucial to conduct such analysis within the framework of larger projects (as you have one). Since it lacks methodological innovation, maybe you can consider submitting it to a journal which is more region-specific.

If you would like to publish in the Atmosphere, maybe you could consider expanding the scope of the methodology and, for instance, also investigate the relationship between temperature and rainfall variability, or defy pre-existing ecological zones and look for patterns on its own.

In its current form, I am asking for major revisions so that you can extend your methodology and go beyond the state-of-the-art.

Author Response

(The authors gave the same response as above.)

Round 2

Reviewer 1 Report

My initial comments for the revision were overall reflected to the revised manuscript. However, I should report that the manuscript still needs significant improvements, particularly on evaluating the adequacy of the rain gauge data, since this propagates directly to the quality of the analysis and the conclusion of the paper. If this paper is going to serve as a “validation paper”, then the gauge data should be carefully quality controlled to feed as the “truth” in assessing satellite products. It would be scientifically unreasonable to evaluate overestimations/underestimations of the satellite estimations over places where there is only a few or even no rain gauge. Given the high spatial variability of the precipitation, it is in principal difficult to estimate precipitation rate by interpolations from gauges that are over 100km away. Otherwise, the paper should be revised as a “comparison paper”, reporting on differences between the gauge and satellite products without discussing overestimations/underestimations (as the gauge would not be considered as the truth here). I would recommend the authors to consider the scope of this study and revise the overall manuscript accordingly.

The authors explained that the major uncertainty of the satellite algorithms are in their “tendency of confusing warm orographic rains and very cold surfaces with precipitation whilst IR algorithms has challenges with warm orographic rains.” A reference should be added to support this. Besides, this is not the only reason for the satellite uncertainties and each product has its own strengths/weaknesses as presented in past studies. The manuscript should add a more detailed and comprehensive explanations on their uncertainties to interpret and discuss their analysis results.

From line 221: Is this part explaining Figure 5? If so, this should be indicated at the beginning of the explanation.

Author Response

please see attachment, thank you!

Reviewer 2 Report

The authors successfully responded to my comments and criticisms. I still believe the work lacks in novelty but should be considered as a proper methodological contribution. Thank you for your efforts.

Author Response

Response to Reviewer 2 Second Round Comments

Point: The authors successfully responded to my comments and criticisms. I still believe the work lacks in novelty but should be considered as a proper methodological contribution. Thank you for your efforts.

Response: The authors would like to use this platform to appreciate you for the very constructive concerns and of course we agree there is always room for improvement and the authors have added more information at the methodology and have provided an improved discussion of the results. Hopefully, you will find much improvement.

Round 3

Reviewer 1 Report

My previous review comments were reflected in the revised manuscript. The paper is considered ready for publication in the journal.

This manuscript is a resubmission of an earlier submission. The following is a list of the peer review reports and author responses from that submission.